# Effect of the Vacuum Impregnation Process on Water Absorption and Nail-Holding Power of Silica Sol-Modified Chinese Fir

**Mengxue Tao** [1,2], **Xia Liu** [1,2] and **Wei Xu** [1,2,*]

1   College of Furnishings and Industrial Design, Nanjing Forestry University, Nanjing 210037, China; taoziya1202@njfu.edu.cn (M.T.); 13770567881@163.com (X.L.)
2   Co-Innovation Center of Efficient Processing and Utilization of Forest Resources, Nanjing 210037, China
*   Correspondence: xuwei@njfu.edu.cn; Tel.:+86-25-8542-7459

**Abstract:** The application of fast-growing Chinese fir (*Cunninghamia lanceolata*) is limited due to low dimensional stability and weak mechanical strength. Silica sol can effectively improve fast-growing fir wood's physical and mechanical properties. In order to clarify the influence of impregnation process parameters on the modification effect, the effect of the vacuum impregnation variants (e.g., pre-vacuum time, pre-vacuum pressure, pressurization time, and pressurization pressure) was discussed using the orthogonal test approach. The optimal modification process was determined by comparing the water absorption and nail-holding power under different modification processes. The range analysis and variance analysis methods were used to study the correlation between process factors and the performance of the modified wood. The results showed that the water absorption and nail-holding power of fast-growing fir wood were significantly improved via vacuum impregnating with silica sol. The optimum process parameters for water absorption and nail-holding power of fast-growing fir as the pre-vacuum time was 30 min, the pre-vacuum pressure was −0.08 MPa, the pressurization time was 3 h, and the pressurization pressure was 1.2 MPa.

**Keywords:** silica sol; impregnation modification process; water absorption; nail-holding power

## 1. Introduction

Wood, with its unique texture and excellent processing properties, has been widely used in the construction and furniture industries [1–3]. However, the forest resources of China are scarce and there is a vital shortage of timber. After the 1970s, plantation forests were planted to satisfy economic building demands and solve the issue of the supply and demand of timber [4]. Chinese fir is one of the main fast-growing forest species, with the advantages of a short growth cycle, uniform material, and less warping [5]. It is widely used in the wood furniture manufacturing field. However, Chinese fir also exhibits the same disadvantages of fast growth. Low density, weak mechanical strength, and short service life limited the use of furniture products [6]. Therefore, there is a need to eliminate the defects of fast-growing wood through modification to improve the range of applications and utilization of Chinese fir.

Chemical modification is one of the traditional methods in wood science. Among them, acetylation is a staple modification method in chemical modification, which reduces the hygroscopicity and improves the durability of the wood through the hydroxyl reaction of acetic anhydride with cell wall polymers [7]. Using different concentrations of acetylation modification on pine wood, Papadopoulos and Pougioula [8] found that acetylation positively affected compressive strength when the wood's weight percent gain was below 16.4%. However, too high a level and duration of modification resulted in a decrease in the compressive strength of the wood. Moreover, acetic acid as a by-product is produced during the acetylation process. The acidity of acetic acid affects the wood's strength, so additional operations are required to remove the by-product after modification [9]. Compared

to acetylation, silica sols can also enhance the dimensional stability and other wood defects through impregnation [10,11]. The impregnation process does not require catalysts. No by-products are produced after modification, and the recovery operation of the treatment solution is more straightforward and faster. Silica sol can effectively penetrate and deposit in the pore structure of wood. The silicon atoms and oxygen atoms are cross-linked with each other to form a stable film attached to the surface of the cell wall, which in turn achieves the purpose of improving the stability of wood [12]. In previous studies, silica sols have been applied to modify the wood's flame retardant and mechanical properties. Pries et al. [13] treated pine sapwood with silica sol and discovered that water absorption and fire retardancy were improved. Fu et al. [14] impregnated dried wood in a silica sol solution, and the silica film formed on the sample's surface effectively increased the wood's hydrophobicity. After applying silica sol to fast-growing Chinese fir, Lou [15] conducted combustion tests on the specimens. The results showed that the flame-retardant properties of the specimens were remarkably improved.

Earlier studies have demonstrated the positive effect of silica sol impregnation on wood properties. However, the alkalinity of silica sol may damage the wood structure during impregnation, resulting in a loss of performance. Adjusting the impregnation process parameters and increasing the silica sol particles can compensate for this negative effect [16]. Pressure also influences the impregnation effect. Silica sol impregnation proceeds slowly at atmospheric pressure and enters the wood mainly via wetting and diffusion [17]. Air in the wood also hampers the impregnation process. Thus, vacuum impregnation is frequently used to ensure effective and rapid impregnation. The air inside the wood is the first to be expelled in a vacuum environment. The modifying liquid penetrates the wood under pressure, allowing more colloidal particles to enter and fill the interior to achieve the purpose of modification [18]. In addition, the amount of impregnation liquid in the wood can be increased to a certain extent by lengthening the vacuum and impregnation times [19]. Although the diffusion of the filler within the wood is difficult to control, the impregnation parameters could be adjusted to increase the filler amount, ensuring the impregnation modification effect. In industrial production, it is necessary to ensure the impregnation modification effect and shorten the time as much as possible so that the impregnation effect and product quality can reach a balance.

In this study, the orthogonal test was used to investigate the effects of pre-vacuum time, pre-vacuum pressure, pressurization time, and pressurization pressure on silica sol impregnation modification, and the best modification process was explored. Superheated steam treatment was applied first as a pre-treatment for Chinese fir to increase the permeability. The WPG was used to show the impregnation effect. The performance was evaluated using two indexes: water absorption and nail-holding power. This experiment aims to explore the optimal process parameters for silica sol impregnation of modified fir. The results provide a theoretical basis for optimizing the industrial silica sol impregnation process.

## 2. Materials and Methods

### 2.1. Test Materials and Equipment

Five-year-old Chinese fir (*Cunninghamia lanceolata*) was sourced from Fujian, China and provided by Yihua Lifestyle Technology Co., Ltd. (Guangdong, China). There were 10 specimens taken from different woods of the same batch. None of the specimens had any apparent defects. The silica sol was purchased from Guangdong Foshan Ke Ning Chemical Co. (Guangdong, China): particle size 10–20 nm, 2.10 mPa·s viscosity, 10 PH value, and 10% solid content. After superheated steam pre-treatment, the Chinese fir samples were divided into nine groups of 15 specimens in each group for weight percent gain measurement. According to GB/T 1927–2021 [20], the dimensions of the Chinese fir were processed as 20 mm × 20 mm × 20 mm (T × R × L) for the water absorption test. There were nine groups, each containing 15 test blocks. Meanwhile, the size of the fir wood tested for nail-holding power was 50 mm × 50 mm × 150 mm (T × R × L) based on GB/T 1927–2022 [21], and eight specimens were used as a group for a total of nine groups.

Under the provisions of YS/T 5002 [22], ordinary low-carbon nails with a length of 45 mm and a top bar diameter of 2.5 mm were used in this test, whose bars are smooth, with no defects such as surface corrosion or missing.

The test equipment used in this experiment mainly includes a programmable constant temperature and humidity box (JY-TH-80, Jie Yang, Guangdong, China), an oven drying machine (DHG-9245A, Yi Heng, Shanghai, China), and a vacuum pressure impregnation tank (RTA-1000, Ri Tong, Shandong, China). SPSS software (International Business Machines Corporation, Armonk, New York, USA), version number IBM SPSS Statistics 25, was used to perform the orthogonal analyses.

### 2.2. Impregnation-Modified Treatment

2.2.1. Steps of Impregnation Treatment

The impregnation process was divided into three main stages (Figure 1):

1.  Superheated steam pre-treatment stage: According to previous research [23], the pre-treatment specimens of Chinese fir were obtained using a superheated steam treatment for 3 h at a temperature of 120 °C to increase their permeability and impregnability. The pretreated specimens were further dried until the wood moisture content ratio was close to 0%, usually within 3%;

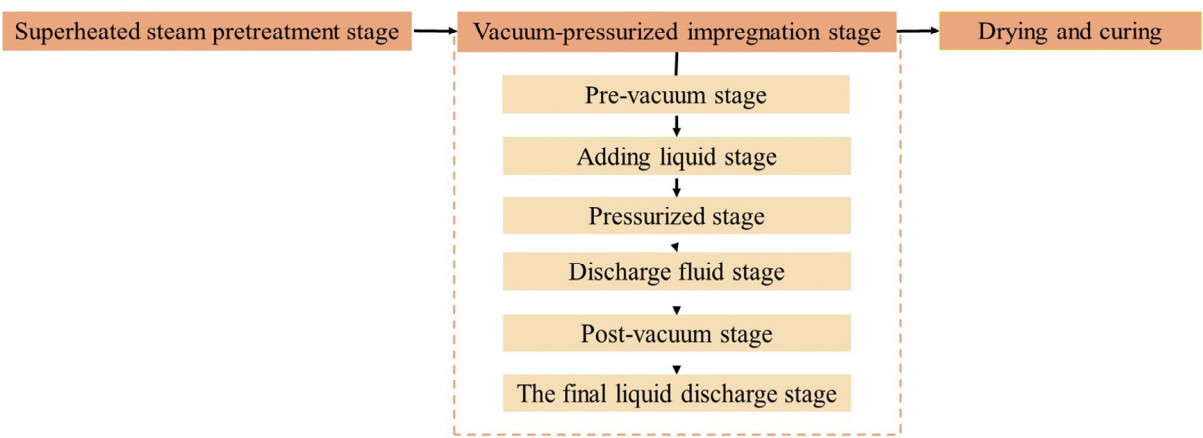

**Figure 1.** Impregnation process of Chinese fir.

2.  Vacuum-pressurized impregnation stage: When the vacuum was pumped, test samples were kept 10–30 mm apart vertically and horizontally to avoid touching the tank so that each sample could be fully impregnated. In addition, ropes were needed to hold the test specimens to prevent the specimens from floating and moving. The test has six stages of the pressure impregnation process: The untreated wood samples were placed in a pressurized impregnation tank. The vacuum pump was turned on to extract the air from the wood samples and maintained for a while after reaching a certain pressure. At this point, the pressure inside the tank was lower than the atmospheric pressure. Then, the impregnating solution entered the impregnation tank and wholly submerged the fir wood. The tank's pressure was increased, and the wood samples were fully impregnated. After pressure removal, the impregnating solution was drained and collected for later use. The vacuum pump was turned on to prevent the impregnating liquid from recoiling out, and the internal pressure was kept at −0.08 MPa for 20 min. Finally, the liquid was drained again, and the modified specimens were removed after a simple rinse;
3.  Drying and curing: The modified wood samples were stacked on wooden strips with a 10–30 mm gap between adjacent timbers. The objective was to ensure ventilation and prevent the wood from becoming moldy. Additionally, heavyweights had to be placed on top of the pile to prevent the deformation of the modified samples. The modified wood samples used a segmented heating and drying method to prevent

cracking and uneven drying. The modified specimens were dried at (25 ± 2) °C, 50 °C, and 90 °C so that the water in the specimens gradually evaporated and the impregnating solution gradually cured.

### 2.2.2. Impregnation Modification Process

A four-factor, three-level orthogonal experiment was designed to investigate the effects of the pre-vacuum and pressurization stages' time and pressure on silica sol impregnation. The four factors in the test contained pre-vacuum time, pre-vacuum pressure, pressurization time, and pressurization pressure, each of which was varied using three levels. Water absorption and nail-holding power were used as evaluation indexes. Table 1 shows the orthogonal test table for the impregnation treatments.

**Table 1.** The orthogonal experiment design of impregnating.

| Number | Pre-Vacuum Time (min) | Pre-Vacuum Pressure (MPa) | Pressurization Time (h) | Pressurization Pressure (MPa) |
|---|---|---|---|---|
| 1 | 10 | −0.04 | 1 | 0.8 |
| 2 | 10 | −0.06 | 2 | 1.0 |
| 3 | 10 | −0.08 | 3 | 1.2 |
| 4 | 20 | −0.04 | 2 | 1.2 |
| 5 | 20 | −0.06 | 3 | 0.8 |
| 6 | 20 | −0.08 | 1 | 1.0 |
| 7 | 30 | −0.04 | 3 | 1.0 |
| 8 | 30 | −0.06 | 1 | 1.2 |
| 9 | 30 | −0.08 | 2 | 0.8 |

### 2.3. Performance Testing

#### 2.3.1. Weight Percent Gain

Weight percent gain (WPG) is one of the essential reference data to reflect the impregnation effect. A higher WPG reflects more impregnation reagents in the wood, improving the impregnation effect. The whole dried fir test samples were weighed. After impregnation was completed, the samples were dried and weighed again. Each set of specimens underwent multiple experiments before the erroneous data were eliminated, and the average result was computed. With an accuracy of within 1%, the weight percent gain is computed as follows:

$$WPG = \frac{W_\tau - W_C}{W_C} \times 100\%$$

where *WPG* is the sample weight percent gain; $W\tau$ is the mass of the sample after impregnation; and $W_C$ is the over-dried mass of the sample before impregnation.

#### 2.3.2. Water Absorption

Wood's water absorption reflects wood's ability to absorb and retain water. It is the ratio of the wood's mass that has absorbed water to saturation (or the mass of water absorbed at a given time) to the mass of completely dry wood.

This test is based on GB/T 1927–2021 [21]. Before the test, the modified wood was placed in the constant temperature and humidity chamber ((20 ± 2) °C for temperature and (65 ± 3)% for humidity) to adjust the moisture content to 12% to avoid bias in the test results due to the difference in moisture content [24]. Furthermore, to ensure the correctness of the test results, the test was repeated for each test piece, and the average value was calculated after excluding the error data.

The formula for calculating water absorption is as follows, accurate to 1%:

$$A = \frac{M - M_0}{M_0} \times 100\%$$

where *A* is the sample water absorption, *M* is the sample's mass after water absorption, and $M_0$ is the over-dried mass of the sample after impregnation.

### 2.3.3. Nail-Holding Power

Before the nail-holding power test, the nails were cleaned and marked 30 mm from the top. Then, perpendicular to the test pieces' radial surface, tangential surface, and two cross surfaces (Figure 2), the steel nails were hammered 5–10 times and nailed to a depth of 30 mm [21]. Then, the nail-holding power was tested using the universal mechanical testing machine (Figure 3). The formula for calculating the nail-holding power is as follows, with an accuracy of 0.1 N/mm:

$$P_{\text{ap}} = \frac{P_{max}}{l}$$

where $P_{\text{ap}}$ is the nail-holding power of the sample, $P_{max}$ is the maximum load, and $l$ represents the nails in the length of the sample.

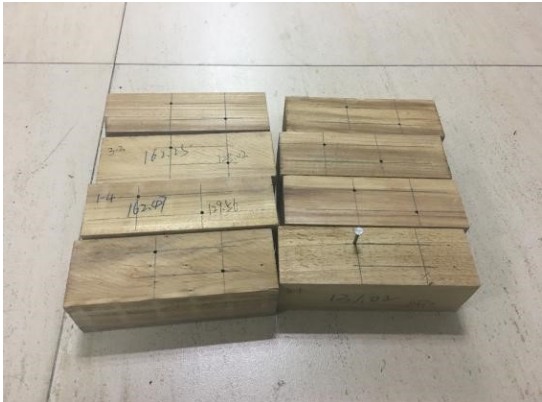

**Figure 2.** Specimen for nail-holding power.

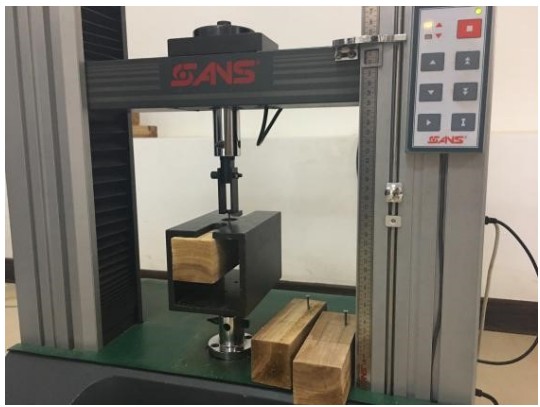

**Figure 3.** Nail-holding power tests using the mechanical testing machine.

Immediately after testing, a block of wood about 50 mm × 50 mm × 10 mm (along the grain direction) was intercepted in the middle of the sample length to measure the samples' moisture content at the present time.

## 3. Results and Discussion

### 3.1. Weight Percent Gain

The experimental data showed an increase in the weight of all the impregnated materials, which indicates that the silica sol impregnated and filled the samples' internal voids (Table 2). The silica sol mainly covered the inner surface of the samples' cell walls at the beginning of impregnation. As the pressurization time increased, the samples' cell cavities were filled with silica sol, and the amount of silica sol adhering to the cell wall decreased. A higher weight of samples indicates that more silica sol was deposited in the cells, and more cell lumens were full of silica sol [25]. The difference in weight percent gain between the groups indicates that the impregnation process has an impact on the

impregnation effect. The WPG of the third and eighth specimen groups exceeds 50% (Figure 4), indicating that more silica sol was deposited in the cell lumen and more cell lumens were full [12,26]. Compared to Group 8, the pressurization time of Group 3 was longer, and the pressurization pressure was higher in both groups. It can be inferred that more silica sol entered the samples' interior under high pressure and continued to be deposited as the pressurization time increased. The fourth group of samples did not show a high weight percent gain at the same pressurization pressure, probably due to the pre-vacuum pressure reduction. The lower pre-vacuum pressure caused the air not to be fully discharged, which prevented silica sol from impregnating and filling the intercellular space. There was a similar performance between the third and eighth groups. The two groups had different values of pre-vacuum pressure, which was slightly lower in Group 8, but the pre-vacuum time was extended accordingly, with less change in the effect on the final weight percent gain.

**Table 2.** The weight percent gain of modified Chinese fir.

| Number | Pre-Vacuum Time (min) | Pre-Vacuum Pressure (MPa) | Pressurization Time (h) | Pressurization Pressure (MPa) | WPG (%) |
|---|---|---|---|---|---|
| 1 | 10 | −0.04 | 1 | 0.8 | 32.7 ± 1.5 |
| 2 | 10 | −0.06 | 2 | 1.0 | 38.8 ± 2.0 |
| 3 | 10 | −0.08 | 3 | 1.2 | 54.4 ± 3.6 |
| 4 | 20 | −0.04 | 2 | 1.2 | 42.7 ± 2.3 |
| 5 | 20 | −0.06 | 3 | 0.8 | 33.1 ± 1.8 |
| 6 | 20 | −0.08 | 1 | 1.0 | 47.6 ± 2.6 |
| 7 | 30 | −0.04 | 3 | 1.0 | 32.0 ± 1.4 |
| 8 | 30 | −0.06 | 1 | 1.2 | 50.3 ± 3.3 |
| 9 | 30 | −0.08 | 2 | 0.8 | 34.8 ± 1.1 |

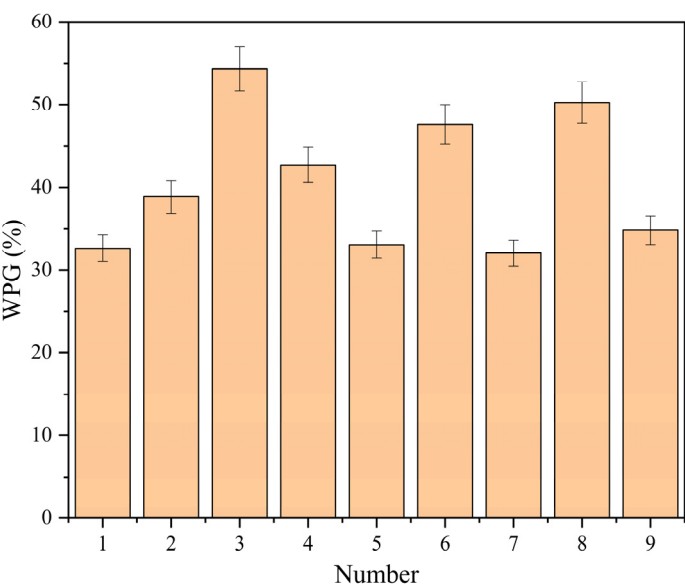

**Figure 4.** The weight percent gain of modified Chinese fir.

*3.2. Water Absorption*

The water absorption curves of the untreated and modified wood samples were plotted according to the measured data, as shown in Figure 5. The water absorption rate was high during the first 24 h of soaking, and it progressively became flat as the soaking period was extended. There was much space to store water inside the thoroughly dried specimens. The water quickly encroached under the pressure difference between the inside and outside. The moisture increase in wood would gradually raise the osmotic pressure, slowing water from entering the wood. Meanwhile, the water absorption rate reduction is

also due to the reduction in internal water storage space. Compared with the untreated fir wood, the impregnated fir wood's water absorption rate was significantly lower. One of the explanations for the phenomenon was that after the impregnation treatment, the silica sol clusters filled part of the wood's voids, blocking some water pathways and occupying water storage space. Thus, the water absorption rate was reduced [27]. In addition, silica molecules bind to -OH in the wood matrix, thereby reducing the wood's binding of moisture and decreasing water absorption [28]. The water absorption curves of the modified materials were in proximity. However, it can still be discovered that the samples in the third, sixth, and eighth groups had lower water absorption rates due to more significant WPG and better impregnation. Such a difference also indicated that the impregnation process affects the material properties. The sixth group of specimens had a lower WPG and better hydrophobicity than the third group. It is possible that part -OH groups on silica sol in the Group 3 specimens absorbed water [12]. Therefore, selecting appropriate process parameters is essential.

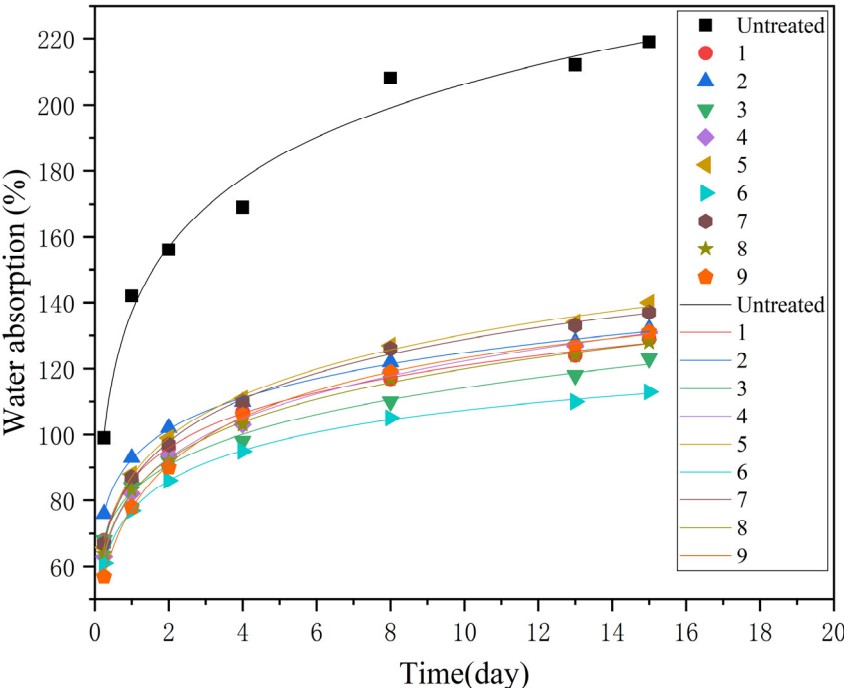

**Figure 5.** Water absorption curve analysis of untreated and modified Chinese fir.

Based on the measured water absorption data of the untreated and modified specimens, the logarithmic function model about water absorption days and water absorption rate was established using regression analysis, as shown in Table 3. The correlation index $R^2$ ranges from 0 to 1. As the value of $R^2$ gets closer to 1, it indicates that the measured value is closer to the regression value. Then, the regression equation has a better fit. The untreated and modified specimens' correlation coefficients $R^2$ were between 0.9826 and 0.9985 (Table 3), indicating that all ten logarithmic functions' regression equations fitted well. Therefore, the ten logarithmic functions can describe the relationship between the water absorption days and the water absorption rate of all test specimens.

As shown in Table 4, within 6 h, the untreated fir samples' water absorption was close to 100%, while the treated samples were below 76% in all cases. Only one group of treated specimens had slightly more than 100% water absorption after 2 days. After 8 days, all treated specimens exceeded 100% water absorption, while untreated materials exceeded 200%. Overall, it was observed that the modified samples' water absorption was significantly lower than untreated.

**Table 3.** Water absorption regression analysis of untreated and modified Chinese fir.

| Number | Pre-Vacuum Time (min) | Pre-Vacuum Pressure (MPa) | Pressurization Time (h) | Pressurization Pressure (MPa) | y = a + b ln x | R$^2$ |
|---|---|---|---|---|---|---|
| Untreated | —— | —— | —— | —— | y = 0.2927ln(x) + 1.3851 | 0.9833 |
| 1 | 10 | −0.04 | 1 | 0.8 | y = 0.1468ln(x) + 0.8702 | 0.9963 |
| 2 | 10 | −0.06 | 2 | 1.0 | y = 0.1353ln(x) + 0.9343 | 0.9942 |
| 3 | 10 | −0.08 | 3 | 1.2 | y = 0.1294ln(x) + 0.8411 | 0.9826 |
| 4 | 20 | −0.04 | 2 | 1.2 | y = 0.1673ln(x) + 0.8341 | 0.9928 |
| 5 | 20 | −0.06 | 3 | 0.8 | y = 0.1795ln(x) + 0.8868 | 0.9939 |
| 6 | 20 | −0.08 | 1 | 1.0 | y = 0.1275ln(x) + 0.7785 | 0.9985 |
| 7 | 30 | −0.04 | 3 | 1.0 | y = 0.1722ln(x) + 0.8853 | 0.9930 |
| 8 | 30 | −0.06 | 1 | 1.2 | y = 0.1573ln(x) + 0.8331 | 0.9939 |
| 9 | 30 | −0.08 | 2 | 0.8 | y = 0.1835ln(x) + 0.8002 | 0.9954 |

**Table 4.** Water absorption of untreated and modified Chinese fir.

| Number | Water Absorption (%) | | | | | | | Time Required for 100% Water Absorption (Day) |
|---|---|---|---|---|---|---|---|---|
| | 1/4 day | 1 day | 2 day | 4 day | 8 day | 13 day | 15 day | |
| Untreated | 99 | 142 | 156 | 169 | 208 | 212 | 219 | 0.2683 |
| 1 | 68 | 86 | 96 | 107 | 117 | 124 | 129 | 2.4210 |
| 2 | 76 | 93 | 102 | 110 | 122 | 128 | 132 | 1.6251 |
| 3 | 68 | 84 | 92 | 98 | 110 | 118 | 123 | 3.4143 |
| 4 | 63 | 82 | 94 | 103 | 119 | 127 | 131 | 2.6956 |
| 5 | 66 | 88 | 99 | 111 | 127 | 134 | 140 | 1.8788 |
| 6 | 61 | 77 | 86 | 95 | 105 | 110 | 113 | 5.6817 |
| 7 | 67 | 87 | 97 | 110 | 126 | 133 | 137 | 1.9466 |
| 8 | 64 | 83 | 92 | 103 | 117 | 124 | 128 | 2.8893 |
| 9 | 57 | 78 | 90 | 106 | 119 | 127 | 131 | 2.9708 |

Table 4 and Figure 5 show that all modified wood's water absorption efficiency decreased. From the fitted curve, the number of days required for the samples to reach 100% water absorption was extrapolated. According to the data, it took 0.2683 days for untreated wood to reach 100% water absorption, while it took 1.6251~5.6817 days for modified wood. The modified wood took 6–21 times longer than the untreated wood to reach 100% water absorption. Moreover, Group 3, with better WPG, took the longest time to reach 100% water absorption, which can also be used to infer how modified wood behaves in long-term moisture absorption. The slopes of the fitted functions for the modified samples are all significantly lower than those of the untreated samples (Table 3 and Figure 5). The study indicated that the modified specimens' water absorption was significantly reduced.

More specifically, range and variance analyses were performed on the modified wood to explore the correlation between process factors and water absorption.

Table 5 is the range analysis of the different days when the modified specimens reached 100% water absorption. Level means the different processing parameters for the four processing factors (i.e., the three levels of pre-vacuum treatment time are 10 min, 20 min, and 30 min). The difference between the maximum and minimum values of the experimental data at various levels is known as the range. A larger range value indicates that the factor has a better positive effect on performance.

The pre-vacuum pressure had the most significant effect on the modified wood's water absorption in Table 5. Moreover, the effects of pressurization time, pre-vacuum time, and pressurization pressure decreased in order. With a lower water absorption rate, wood products are more durable and have better dimensional stability. Modified wood had the best hydrophobicity and took the longest to reach 100% water absorption when the pre-vacuum time was 10 min, the pre-vacuum pressure was −0.06 MPa, the pressurization time was 3 h, and the pressurization pressure was 0.8 MPa.

**Table 5.** Range analysis of modified Chinese fir on water absorption.

| Properties | Level | Pre-Vacuum Time (min) | Pre-Vacuum Pressure (MPa) | Pressurization Time (h) | Pressurization Pressure (MPa) |
|---|---|---|---|---|---|
| 100% water absorption rate (%) | 1 | 2.487 | 2.354 | 3.664 | 2.424 |
| | 2 | 3.419 | 2.131 | 2.431 | 3.084 |
| | 3 | 2.602 | 4.022 | 2.413 | 3.000 |
| | Range (R) | 0.932 | 1.891 | 1.251 | 0.660 |

Table 6 shows the variance analysis and significance test for the effect of variable factors on modified wood's water absorption. The significance of all four factors was more than 0.05, indicating that none significantly impacted the water absorption of the modified specimens.

**Table 6.** Variance analysis and significance test of designed parameters on water absorption of modified Chinese fir.

| Factor | Sum of Squares | DF | F Value | F Critical Value | Sig |
|---|---|---|---|---|---|
| Pre-Vacuum Time | 1.548 | 2 | 1.995 | 19 | >0.05 |
| Pre-Vacuum Pressure | 6.408 | 2 | 8.258 | 19 | >0.05 |
| Pressurization Time | 3.086 | 2 | 3.977 | 19 | >0.05 |
| Pressurization Pressure | 0.776 | 2 | 1.000 | 19 | >0.05 |
| Error | 0.78 | | | | |

Note: Significance level $\alpha = 0.05$.

### 3.3. Nail-Holding Power

Figure 6 and Table 7 show the analysis of the nail-holding power for the untreated and modified specimens.

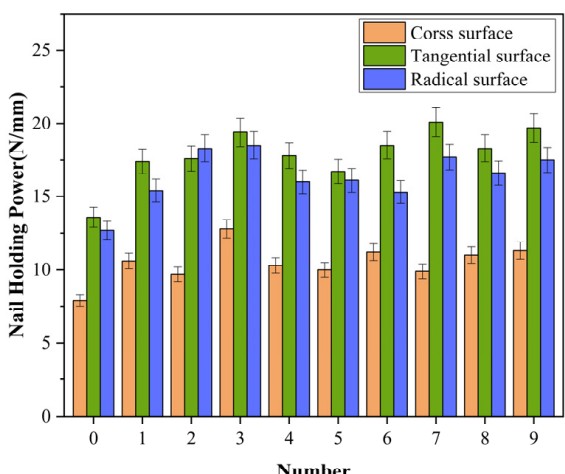

**Figure 6.** The nail-holding power of untreated and modified Chinese fir.

As shown in Table 7, cross surfaces' nail-holding power of untreated fir wood was about 7.9 N/mm and increased by 23% to 62% about modified wood, reaching 9.7 N/mm at the minimum and 12.8 N/mm at the maximum. For the tangential surface, untreated wood had a nail-holding power of 13.6 N/mm, whereas modified wood had a nail-holding power of 16.7–20.1 N/mm, with an increase of 23%–48%. From the radial surface of impregnated wood, the nail-holding power increased from 12.7 N/mm to 15.3–18.5 N/mm, which increased by 20%–45%. The results demonstrated that all three surfaces' nail-holding power was greatly enhanced after impregnation modification.

**Table 7.** The nail-holding power analysis of untreated and modified Chinese fir.

| Number | Pre-Vacuum Time (min) | Pre-Vacuum Pressure (MPa) | Pressurization Time (h) | Pressurization Pressure (MPa) | Moisture Content (%) | Nail-Holding power at 12% Moisture Content (N/mm) | | |
|---|---|---|---|---|---|---|---|---|
| | | | | | | Cross Surface | Tangential Surface | Radial Surface |
| 0 | | Untreated | | | 10.2 | 7.9 ± 2.3 | 13.6 ± 3.8 | 12.7 ± 3.5 |
| 1 | 10 | −0.04 | 1 | 0.8 | 11.6 | 10.6 ± 2.1 | 17.4 ± 3.9 | 15.4 ± 3.3 |
| 2 | 10 | −0.06 | 2 | 1.0 | 12.5 | 9.7 ± 2.1 | 17.6 ± 4.1 | 18.3 ± 4.0 |
| 3 | 10 | −0.08 | 3 | 1.2 | 13.0 | 12.8 ± 2.5 | 19.4 ± 4.2 | 18.5 ± 4.2 |
| 4 | 20 | −0.04 | 2 | 1.2 | 11.8 | 10.3 ± 2.1 | 17.8 ± 4.0 | 16.0 ± 3.4 |
| 5 | 20 | −0.06 | 3 | 0.8 | 10.6 | 10.0 ± 1.9 | 16.7 ± 3.8 | 16.1 ± 3.2 |
| 6 | 20 | −0.08 | 1 | 1.0 | 11.2 | 11.2 ± 2.6 | 18.5 ± 4.2 | 15.3 ± 3.3 |
| 7 | 30 | −0.04 | 3 | 1.0 | 12.5 | 9.9 ± 2.0 | 20.1 ± 4.3 | 17.7 ± 3.8 |
| 8 | 30 | −0.06 | 1 | 1.2 | 10.9 | 11.0 ± 2.2 | 18.3 ± 3.2 | 16.6 ± 3.0 |
| 9 | 30 | −0.08 | 2 | 0.8 | 12.5 | 11.3 ± 2.3 | 19.7 ± 4.4 | 17.5 ± 3.3 |

Among the three fir wood sections, the tangential surface has the best nail-holding power, while the cross surface is the lowest. After the nail vertically passes through the cell wall, it will squeeze the wood fiber axially, with a tight fit between the nail body and the wood fiber. However, when nailed along the fiber direction, the cell wall will be squeezed into the surrounding cell cavity. As a result, the friction between the nail and the wood fibers is less, and the nail-holding power is decreased [29]. The nail-holding power of all three surfaces increased to different degrees after the modification treatment (Figure 6). The improvement in nail-holding power may be attributed to increased wood density caused by silica sol filling [30]. The friction between the nail body and the wood fibers increases with density. After different impregnation treatments of all group specimens, the nail-holding power on three surfaces also showed distinctions. The specimens of Group 3 with the best WPG showed better nail-holding power on cross and radial surfaces. Meanwhile, the nail-holding power of the tangential surface also increased by 42.6%. When the nail was driven into the sample, the wood fibers squeezed the nail body, creating friction. Moreover, after impregnation, the voids within the sample were filled with silica sol clusters, which likewise gave the nail a squeezing force [12,25,26]. When the nail was pulled out, the nail needed to overcome the combined force from the wood fibers and the silica gel mass. With better WPG, more silica sol clusters increased the friction between the wood and the nail body, and the sample achieved a better nail grip. The specimens of Group 7 showed a tremendous increase in tangential surface nail-holding power but a weaker rise on the cross surface. Group 7 samples had a lower WPG, suggesting that only a small number of cell lumens were full of silica sol, and more silica sol clusters adhered to the cell walls [25]. The nail passes through more cell walls per unit length when nailed in the vertical chord section. The area where the nail body contacted the wood cell wall and silica sol was larger, resulting in greater friction and enhanced nail-holding power. The nail-holding power was enhanced with the increase in silica sol.

Similarly, range analysis and variance analysis for modified fir showed the effect of process factors on nail-holding power.

The range analysis of the nail-holding power of modified Chinese fir is shown in Table 8.

Among the factors affecting nail-holding power, pre-vacuum pressure had the most excellent effect on the cross surface because of the enormous value of the range. Pressurization pressure had the following highest effect on nail-holding power. In contrast, smaller values of the range between pre-vacuum time and pressurization time demonstrated weaker impacts on nail-holding power. The modified wood's highest cross-surface nail-holding power was obtained when the pre-vacuum time was 10 min, pre-vacuum pressure was −0.08 MPa, pressurization time was 1 h, and pressurization pressure was 1.2 MPa.

**Table 8.** Range analysis of variables on nail-holding power of modified Chinese fir.

| Properties | Level | Pre-Vacuum Time (min) | Pre-Vacuum Pressure (MPa) | Pressurization Time (h) | Pressurization Pressure (MPa) |
|---|---|---|---|---|---|
| Cross-surface nail-holding power (N/mm) | 1 | 11.033 | 10.267 | 10.933 | 10.633 |
| | 2 | 10.500 | 10.233 | 10.433 | 10.267 |
| | 3 | 10.733 | 11.767 | 10.900 | 11.367 |
| | Range (R) | 0.533 | 1.534 | 0.500 | 1.100 |
| Tangential surface nail-holding power (N/mm) | 1 | 18.133 | 18.433 | 18.067 | 17.933 |
| | 2 | 17.667 | 17.533 | 18.367 | 18.733 |
| | 3 | 19.367 | 19.200 | 18.733 | 18.500 |
| | Range (R) | 1.700 | 1.667 | 0.666 | 0.800 |
| Radial surface nail-holding power (N/mm) | 1 | 17.400 | 16.367 | 15.767 | 16.333 |
| | 2 | 15.800 | 17.000 | 17.267 | 17.100 |
| | 3 | 17.267 | 17.100 | 17.433 | 17.033 |
| | Range (R) | 1.600 | 0.733 | 1.666 | 0.767 |

For the tangential surface, the most range of the pre-vacuum time meant a significant influence on nail-holding power. The pre-vacuum pressure exerted a slightly weaker effect. The ranges of both pressurization pressure and pressurization time were significantly lower than the pre-vacuum and pre-vacuum pressure. As a result, both the pressurization pressure and the pressurization time had a lower effect on the tangential surface's holding power. When the pre-vacuum time was 30 min, the pre-vacuum pressure was $-0.08$ MPa, the pressurization time was 3 h, and the pressurization pressure was 1 MPa, the tangential surface's nail-holding power of the modified specimens was the highest.

For the nail-holding power of the radial surface, the range between the pre-vacuum time and the pressurization time was close to the same value, which had a more extensive influence among the four influencing factors. The ranges of pressurization pressure and pre-vacuum pressure were small and almost equal, so the influence of the two was also weak. The optimal impregnation process parameters were pre-vacuum pressure $-0.08$ MPa for 10 min and impregnation at pressure 1 MPa for 3 h.

The optimum impregnation process parameters differed for the three wood surfaces. In actual production, the furniture requires higher nail-holding power on the radial and tangential surfaces [31]. Therefore, the process parameters should be selected to achieve the best nail-holding power on radial and tangential surfaces while ensuring better cross-surface nail-holding power. The optimal pre-vacuum pressure, pressurization time, and pressurization pressure were the same for the radial and tangential surfaces. In contrast, the pre-vacuum time significantly influenced the tangential surface's nail-holding power. Thus, the optimum modified specimens' nail-holding power was achieved at a pre-vacuum time of 30 min, a pre-vacuum pressure of $-0.08$ MPa, a pressurization time of 3 h, and a pressurization pressure of 1 MPa.

Table 9 shows the analysis of variance and significance test for the nail-holding power of modified Chinese fir. For the cross surface, pre-vacuum pressure had some impact on nail-holding power, as evidenced by the significance of pre-vacuum pressure being less than 0.05. On the other hand, pre-vacuum time, pressurization time, and pressurization pressure all had significances greater than 0.05, indicating that they had no discernible effects on the nail-holding power. Moreover, the significance of all four factors was more than 0.05 for the radial and tangential surfaces, demonstrating that the four parameters did not significantly affect the modified wood's nail-holding power.

**Table 9.** Variance analysis and significance test of nail-holding power of modified Chinese fir.

| Properties | Factor | | Sum of Squares | DF | F Value | F Critical Value | Sig |
|---|---|---|---|---|---|---|---|
| Cross surface | Pre-Vacuum | Time | 0.429 | 2 | 0.915 | 19 | >0.05 |
| | | Pressure | 4.602 | 2 | 9.812 | 19 | <0.05 |
| | Pressurization | Time | 0.469 | 2 | 1.000 | 19 | >0.05 |
| | | Pressure | 1.882 | 2 | 4.013 | 19 | >0.05 |
| | Error | | 0.47 | 2 | | | |
| Tangential surface | Pre-Vacuum | Time | 4.629 | 2 | 6.919 | 19 | >0.05 |
| | | Pressure | 4.176 | 2 | 6.242 | 19 | >0.05 |
| | Pressurization | Time | 0.669 | 2 | 1.000 | 19 | >0.05 |
| | | Pressure | 1.016 | 2 | 1.519 | 19 | >0.05 |
| | Error | | 0.67 | 2 | | | |
| Radial surface | Pre-Vacuum | Time | 4.729 | 2 | 4.983 | 19 | >0.05 |
| | | Pressure | 0.949 | 2 | 1.000 | 19 | >0.05 |
| | Pressurization | Time | 5.056 | 2 | 5.328 | 19 | >0.05 |
| | | Pressure | 1.082 | 2 | 1.140 | 19 | >0.05 |
| | error | | 0.95 | 2 | | | |

Note: Significance level $\alpha = 0.05$.

### 3.4. Synthesize and Analyze

The four factors were ranked in a descending order of range values to further understand each process factor's impact on water absorption and nail-holding power (Tables 5 and 8). It showed that pre-vacuum time had the most significant impact on tangential surface nail-holding power, followed by radial surface nail-holding power. Therefore, the pre-vacuum time of 30 min, which produces superior tangential and radial surface nail-holding power, should be preferred. Pre-vacuum pressure is essential for both water absorption and tangential nail-holding power. Consequently, pre-vacuum pressures of −0.06 and −0.08 were given priority. By comparing the data with actual measurements (Tables 4 and 7), when the pre-vacuum pressure was −0.08, the samples exhibited better water absorption and nail-holding power. Pressurization time mainly affected radial surface nail-holding power and water absorption, which performed best when the pressurization time was 3 h. Among the four factors, pressurization pressure had a weaker effect on water absorption and nail-holding power than the other. Pressurization pressure had a decreasing effect on cross-surface, tangential surface, radial surface nail-holding power, and water absorption order. A pressurization pressure of 1.2 MPa was chosen to prioritize the cross-nail-holding power.

In summary, when the pre-vacuum time was 30 min, the pre-vacuum pressure was −0.08, the pressurization time was 3 h, and the pressurization pressure was 1.2 MPa, the modified wood with low water absorption and high nail-holding power could be obtained.

### 4. Conclusions

In the current research, the heat-treated fast-growing fir wood was impregnated with silica sol via vacuum impregnation. The suitable parameters for silica sol impregnation were also discussed to provide a reference for the silica sol impregnation of Chinese fir. The modified samples' weight percent gain, water absorption, and nail-holding power were analyzed. The main conclusions of this study were as follows:

- Compared to the untreated wood, the water absorption of modified specimens was significantly reduced in all the different process-modified treatments. There were some variations in water absorption between specimens treated using different processes.

Pre-vacuum pressure, pressurization time, pre-vacuum time, and pressurization pressure had a decreasing effect on the water absorption of the modified wood in order;

- After impregnation modification, the nail-holding power was significantly improved on all three surfaces of the test specimens. Moreover, the nail-holding power of tangential and radial surfaces was higher than the cross surface;

- Considering the impact of the four factors on performance, the best procedure to achieve both lower water absorption and better nail-holding power was 30 min of pre-vacuum time, −0.08 MPa of pre-vacuum pressure, 3 h of pressurization time, and 1.2 MPa of pressurization pressure.

**Author Contributions:** Writing—review and editing, M.T.; writing—original draft, X.L.; supervision, W.X. All authors have read and agreed to the published version of the manuscript.

**Funding:** This research was funded by the Qinglan Project of the Jiangsu Province of China.

**Data Availability Statement:** The generated data are already contained in the text. The datasets generated and/or analyzed during the current study are not publicly available because data processing involves another submission but can be obtained from the corresponding author according to reasonable requirements.

**Acknowledgments:** We thank the teachers of the Advanced Analysis and Testing Center of Nanjing Forestry University for their help in instrument testing, and we are grateful to the reviewers and editors for their valuable time and suggestions for improving the quality of this paper.

**Conflicts of Interest:** The authors declare no conflict of interest.

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
