# Peer review of "Effect of the Vacuum Impregnation Process on Water Absorption and Nail-Holding Power of Silica Sol-Modified Chinese Fir"

_forests, doi:10.3390/f15020270_

Round 1
Reviewer 1 Report
Comments and Suggestions for Authors
The manuscript is well written with deep discussion. Detail comments and corrections are provided in the manuscript. It is hoped that the authors could recommend the appropriate treatment to obtain the best improvement both for nail holding power and water absorption.
Need consistency in the references writing.

The English is good, but please read again carefully before re-submission.
Author Response
lease see the attachment

Reviewer 2 Report
Comments and Suggestions for Authors
Line 146 to 147 - experimental design must be improved. Impregnation time and Impregnation pressure are not clear. Example, where is 1 minute impregnation time for 1.0 MPa and 1.2 MPa (Impregnation pressure. Same as 2 minutes impregnation time and where is the 0.8 MPa and 1.2 MPa (impregnation pressure. Same to 20 minutes pre-vacuum time and 30 minutes vacuum time?.
Line 202 to 204 - It is speculative only. It needs at least scanning electron microscope (SEM) figure to prove it.
Line 226 to 227 - The title in reference 25 is not related to the statement in the text.
Line 233 to 234 - It is speculative only without at least SEM figure to prove it. If no SEM figure, you need to support your finding with any related reference (literature review).
Table 5 - is not understand and really confuse. Please explain about the Table input so that it is more understand.
Line 307 to 310 - your statement is only speculative and without prove with at least SEM Figure. You need to find any literature or reference to support your opinion.
Reviewer 3 Report
Comments and Suggestions for Authors
The contribution is readable and understandable. In spite of understand ability of the contribution, it contains some imperfections, which should be eliminated or more explained.
Line 11: …Chinese fir is limited by low dimensional stability… Line 29 …wood with excellent physical properties… (Such sentences confuse the reader.)
Lines 22, 24 …: Negative sign for the pressure is suspicious and not usual.
Line 45: introduce abbreviation WPG; Line 85 abbreviate wood percentage gain
Line 94: use only SI unit.
Line 165: What is difference between moisture content (Line 180) and water absorption A? Are Wc and M0 equal? Explain more.
Line 263: The units of the properties should be added.
Round 2
Reviewer 2 Report
Comments and Suggestions for Authors
After the correction, this paper is acceptable for publication.